# Educating Youth to Civic Engagement for Social Justice: Evaluation of a Secondary School Project

**DOI:** 10.3390/bs13080650

**Published:** 2023-08-03

**Authors:** Mara Martini, Chiara Rollero, Marco Rizzo, Sabrina Di Carlo, Norma De Piccoli, Angela Fedi

**Affiliations:** 1Department of Psychology, University of Turin, 10124 Torino, Italy; chiara.rollero@unito.it (C.R.); marco.rizzo@unito.it (M.R.); norma.depiccoli@unito.it (N.D.P.); angela.fedi@unito.it (A.F.); 2“Spostiamo mari e monti” (“Let’s Move Seas and Mountains”) Social Promotion Association, 10145 Torino, Italy; sab.dicarlo@gmail.com

**Keywords:** civic engagement, trust, community psychology, social justice

## Abstract

The positive effects of youth civic engagement can be felt both at the individual level (e.g., better emotional regulation, a greater sense of empowerment) and at the community level (e.g., a greater likelihood of participation in civic and political activities). They may also be a protective factor for at-risk youth in the short and long term and a valuable element for positive identity development in general. The purpose of this longitudinal study was to assess the impact of an educational intervention implemented in secondary schools to promote youth civic engagement (*N* = 508 at Time 1, *N* = 116 at Time 2). The study is divided into two parts: first, it examines the changes stimulated by the project, and second, it uses a path analysis model to explain the intention to participate. Results show that after participation, hostile and benevolent sexism, classic and modern ethnic prejudice, and social dominance orientation decreased, while trust in institutions increased. In addition, the path analysis showed that policy control, social trust, and civic engagement increased the intention of civic engagement at time T1. Despite some limitations, this study may provide useful guidance for those designing and implementing civic education interventions for young people.

## 1. Introduction

Civic engagement is defined by several scholars [1,2,3] and refers to various practical manifestations “from individual voluntarism to organizational involvement to electoral participation” (p. 3, [1]). It can be declined differently depending on the context in which it is developed, as it is closely tied to the local community [1]. However, some authors proposed a broader definition, namely “individual and collective activities intended to identify and address issues of public concern and enhance the well-being of one community and of the society” (p. 1830, [3]) [4,5]. The positive effects of civic engagement are one of the reasons this construct has received so much attention from scholars. These effects can be observed at both the individual and community levels [3,6,7], and they can vary by age group. Although several papers focus on civic engagement among older adults [1,7], one of the main target groups for analysis and consideration is youth. The potential positive impacts of youth civic engagement are numerous, both short- and long-term [7]. At the individual level, some studies show that higher levels of civic engagement are associated with better emotion regulation and identity reflection, a lower rate of depressive symptoms and problem behaviors [3,8,9,10], as well as greater leadership development and a sense of empowerment [3,7,9]. At the community level, individuals with higher levels of civic engagement showed a greater likelihood of participating in civic, political, association, and activism activities [3,7]. Such participation creates opportunities for interaction with both peers and civically engaged adults and provides opportunities to learn about positive role models, socialize with civic values, and develop relationships based on trust and (civic) responsibility [7,11].

Although there are not many longitudinal studies along these lines, some work demonstrates that civic engagement in adolescence can have positive effects in adulthood as well, both in terms of better emotional regulation and a greater sense of empowerment [3] and a higher likelihood of engaging in civic activities as an adult [3,7,12].

Given all these positive effects of youth civic engagement, several scholars have analyzed the dimensions that can contribute to its development. For example, there is evidence of the crucial role of the cohesion of the social environment to which young people belong and their personal connectedness to that social context, which provides both positive role models and the opportunity to learn, observe, and experiment in a protected setting [7,13].

Positive role models and the opportunity to learn in a protected context are typical of educational initiatives. Finally, Lenzi et al. (p. 45, [7]) define civic engagement as “the feeling of responsibility towards the common good, the actions aimed at solving community issues and improving the well-being of its members, and the competences required to participate in civic life”. It points out that civic engagement is also a fruit of the competencies required to participate in civic actions as well as the perception of having these skills and feeling ready to participate. Civic engagement can thus be learned, and educational proposals can promote its development.

### 1.1. Civic Engagement between Social and Individual Dimensions

In order to plan educational activities that can activate and promote youth civic engagement, it is crucial to focus on factors that support the development of civic engagement [11,14]. Among other factors, scholars have identified close ties and trusting relationships with the local social environment or neighborhood [7], political involvement and personal agency, and the ability to exert influence in social relationships [15]. In other words, the development and maintenance of civic engagement are driven by two types of factors: social and individual.

The social dimension is represented by youth’s relational context, the environment that provides opportunities for interactions and involvement but also for relationships with both adults and peers. Such relationships provide young people with role models and values through supportive relationships with someone who is nearby and easy to reach [13]. This type of context can be the local community where youth live as well as the educational context, which can provide positive models, values, and a non-judgmental learning situation. This allows youth to experience a sense of safety and develop social trust [16]. Social trust is defined as “the belief that most people can be trusted and are basically honest” (p. 219, [16] and [17]). For the reason that it contributes to the development of strong bonds with others in the community, it also creates the conditions for maintaining a willingness to put oneself on the line and commit in turn. Social trust is associated with greater engagement in the local community (or educational context) and an intention to actively contribute to social well-being [16]. In other terms, a supportive social context contributes to the development of a frame in which civic engagement can be solicited and expressed through young people’s behaviors of voluntarism, participation in school and associations, activities, protest and political manifestations, donations, and petitions [2].

Thus, trusting social relationships with adults and peers creates the framework in which young people feel comfortable engaging in community and political matters. However, this is not enough. Developing a willingness to engage in social matters also requires a certain level of institutional trust, which is useful in recognizing that their engagement will not be “a waste of time” (p. 14, [18]). Indeed, their willingness to engage may be based on a belief that institutions will do their part to address social issues.

On the other side, young people need to feel that they have the ability to intervene and to make a difference if they are active and engaged. According to several authors [19,20,21,22,23], civic engagement is sustained by both an attitude of critical analysis of the social situation and sociopolitical control. According to Zimmermann and Zahniser [24], sociopolitical control is expressed in terms of policy control and leadership competence, i.e., on the one hand, the belief that one has sufficient skills to act effectively in social and political contexts and, on the other hand, the belief that one has sufficient influence over others to direct them toward a social goal. These social and personal dimensions have been widely explored in relation to civic engagement, e.g., [13,21].

However, when planning an educational intervention, other dimensions strictly related to the peculiar topic of youth engagement should also be considered. In particular, prejudice and legitimizing beliefs that may discourage engagement can play a special role. Some researchers, e.g., [25,26], argue that reducing prejudice and collective action approaches to social change are contradictory and that in historically unequal societies, intergroup contact can perpetuate injustice by undermining collective action. In contrast, other studies suggest that intergroup contact can unite social groups in action against social injustice and that intergroup contact and collective action can jointly contribute to social change [27].

Thus, the relationship between intergroup contact and collective action is multifaceted and complex. Here, we limit ourselves to considering the possibility that real or imagined, e.g., [28], contact between members of advantaged and disadvantaged groups may reduce the willingness of the advantaged to maintain privileges that result from an unequal situation. As well, this contact can foster the legitimacy of engagement by members of disadvantaged groups [29]. Furthermore, educational interventions can help members of advantaged groups recognize prejudices and related inequalities, encouraging them to engage in critical analysis and social identification, which in turn promote collective action [30,31].

Research on civic engagement also shows that people are more willing to engage if they recognize inequalities: being victimized, seeing oneself as capable of collectively redressing disadvantage, and feeling outraged and angry about disadvantage are robust predictors of active engagement [30,32]. Particularly, moral outrage toward the general system and moral beliefs can connect people beyond their affiliations and motivate them to take collective action [33]. In this perspective, “engagement can be considered a form of action-oriented coping with perceived injustice” (p. 179, [34]). With Acar and Ulug (p. 176, [35]), it can also be argued “that collective action experiences are an extremely powerful way to not only reduce intergroup prejudice but also to take steps toward social justice”.

The perpetuation of inequalities is justified and reproduced through prejudices and individual or institutional forms of discrimination. It can be reinforced by legitimizing myths, that is, values, attitudes, beliefs, norms, and cultural ideologies that act in the direction of promoting or mitigating inequalities [36]. Among these ideologies, Social Dominance Orientation (SDO [37]) is defined as “the extent that one desires that one’s ingroup dominance and be superior to outgroups” (p. 742), an expression of belief in social inequalities. People who tolerate or prefer group-based dominance would be less motivated to engage in collective behaviors to promote social justice [38,39,40]. Educational interventions can address these dimensions directly as discussion topics or indirectly through experiences or activities that support critical reflection.

### 1.2. Education for Civic Engagement

Many studies support the usefulness of educational initiatives for developing civic engagement among youth [41,42,43,44,45]. Considering the various factors that can promote youth civic engagement, Kahen and Sporte [46] demonstrated that civic education initiatives can promote civic engagement even more than academic achievement support and neighborhood and family civic context. Childhood experiences of civic engagement in the family and the norms and strength of social relationships in the community context should not be ignored, but the findings of Kahen and Sporte [46] suggest that school-based and educational initiatives can foster the development of youth civic engagement even when the initial social context is not favorable. This is also encouraging because school is the only institution that can reach all young people in the country [13,47]. School can provide youth with various opportunities for civic learning, both within the curriculum and in extracurricular activities. Youniss [48] identified three types of initiatives that can meet young people’s civic education needs: opportunities to participate in extracurricular government and civic initiatives, involvement in service (such as volunteering) that can foster engagement and identification with a community, and facilitating classroom discussion and debate about critical issues. Indeed, these activities provide opportunities to develop knowledge and reflection on civic and political issues, learn about positive models of behavior from peers and adults, and belong to a community, which are key elements in developing civic engagement [7]. In addition, Pasek and colleagues [49] found that these types of initiatives can promote long-term civic and political participation.

### 1.3. The Project

The present research aimed to evaluate the impact of an educational project designed to promote civic engagement. The project was born from the collaboration between some associations that have been organizing cultural projects on the theme of historical memory in Europe and which have the activation of youth citizenship as their mission. The university collaborated to evaluate the impact of the project. The education project proposed a training course for some high school classes in the northern Italian regions. It focused on the possibilities for each individual to act and introduced the concepts of righteousness and personal responsibility by following the stories of men and women who are the storyline of a journey through history. 

The proposed educational methodology was based on the principles of non-formal education and on the tools of peer education, learning by doing in schools, and an experiential community journey. The project was developed during the 2019–2020 school year between November (T1) and March (T2) in the schools that participated in the project. It consisted of two main phases: a. an in-class peer education training; b. an experiential community journey to some places associated with European memory.

All students in the involved classes participated in the peer education training (phase a), which was led by a group of peer educators who had previously worked on the topics of memory, prejudice, and civic engagement. Each class involved in the project participated in at least three meetings, each lasting two hours and covering a variety of topics. The first meeting focused on a basic chronology of historical events that would have been explored in depth during the journey, as well as propaganda and enemy image building through a workshop activity. The second meeting focused on pressing contemporary issues (minority rights, migration, the environment, and European citizenship). The memory of the Shoah was chosen because it can be a magnifying glass that allows us to perceive the emergence of hatred in different situations in order to try to prevent discrimination, prejudice, and new genocides. During the third meeting, a selection of educational, instructional, and site-bibliographic materials were provided in preparation for the journey.

After the peer education training, some classes participated in an eight-day community journey (phase b) to the places associated with European memory (Krakow, Prague, Berlin, and Budapest). Travel is a typically human experience and can be an important event in creating youth identity and sociality by laying the foundation of community. The choice of a slow mode of transportation (i.e., train) and the many hours of travel aim to create the right distance from daily reality and foster the growth of the traveling community. The journey provides on-the-ground learning that triggers a positive relational dynamic that promotes peer group activity and engagement. This is consistent with the role of the adults—classroom teachers and peer educator coordinators—as supervisors and facilitators of interaction among the youth; they are responsible for training the peer educators, organizing the itineraries, and trying to support this temporary “traveling community”. After the journey, activities were even carried out to tell and give back to the other schoolmates and the citizenship of the lived experience. The storytelling was curated by the participants, who used a variety of languages and tools (e.g., video, photography, theater, and music) to express their emotions and develop their self-consciousness while sharing their experience.

### 1.4. The Current Study: Objectives and Hypotheses 

Thus, to evaluate the impact of the described education project, longitudinal research was carried out. Based on the literature previously reported, we aimed to investigate three main aspects.

The first objective was to evaluate the effect of participation in the project on attitudes and prejudices toward stigmatized groups. Specifically, to test the first aim, we compared the participants’ endorsement of the most widespread Prejudices (i.e., toward women, the LGBT population, and immigrants) and ideologies (i.e., Social Dominance Orientation) before and after the involvement in the project. We hypothesized that:

**H1.** 
*The involvement in the project would reduce both prejudices and ideology related to Social Dominance Orientation.*


The second objective of the work was to evaluate the effect of participation in the project on the constructs promoting civic engagement. To test the second aim, we compared the participants scores on constructs sustaining participation (i.e., Leadership Competence, Policy Control, Social Trust, and Trust in Institutions) before and after their involvement in the project. We expected that: 

**H2.** 
*Each construct’s sustaining participation increases after involvement in the project.*


The third aim of the paper was to investigate which constructs are associated with Intention of and actual Civic Engagement after the end of the project. Thus, we tested a path based on the literature review above described. We hypothesized that:
**H3.** *Leadership Competence, Policy Control, Social Trust, Trust in Institutions, and Civic Engagement at T1 would be positively associated with Intention of Civic Engagement, which in turn would positively influence Civic Engagement at T2.*

## 2. Materials and Methods

### 2.1. Participants and Procedure 

Before being involved in the project (T1) and after the final activity (T2), all students in the classes involved in the education project were invited to fill out an online questionnaire. The link from the posting led participants to a secure, anonymous online questionnaire. Before beginning the study, participants read an informed consent form. The informed consent form was also signed by parents for participants under 18. In line with the Declaration of Helsinki, participants were also informed that their participation was voluntary and that they could stop the study at any time. The completion of the questionnaire took around 30 min. No compensation was paid for participation in the study. 

Participants were 508 (women = 60.8%, men = 37%, other = 2.2%) high school students living in the North of Italy. 69.6% of them were 18 years old, 17.1% were 17, 7.8% were 19, and the others were 16. Most of them (97.6%) lived with their family, whereas 2.4% lived with their partner or friends. The 56.9% had one sibling, the 20.5% were only son/daughter, and the others had two or more siblings. The first wave of data collection (Time 1—T1) took place before the beginning of the project. The second wave of data collection took place at the end of the project. At Time 2 (T2), respondents were 116 (women = 61.3%, men = 38.7%). T1 and T2 data were matched with an anonymized alphanumeric code.

### 2.2. Measures

In both waves, participants were asked to complete a questionnaire that included the measures described below.

The short version of the Ambivalent Sexism Inventory (ASI) [50,51] includes 6 items (α at T1 = 0.85, α at T2 = 0.87) measuring Hostile Sexism toward women (e.g., “Women seek to gain power by getting control over men”) and 6 items (α at T1 = 0.83, α at T2 = 0.84) measuring Benevolent Sexism toward women (e.g., “Many women have a quality of purity that few men possess”). The items were rated on a 6-point point Likert-type scale ranging from “strongly disagree” (0) to “strongly agree” (5).

The LGBT Prejudice Scale [52] includes 3 items (e.g., “I would be ashamed if one member of my family were gay or lesbian”; α at T1 = 0.69, α at T2 = 0.72). The items were rated on a 6-point point Likert-type scale ranging from “strongly agree” (1) to “strongly disagree” (5).

The Classical and Modern Ethnic Prejudice Scale [53,54] contains 15 items grouped together into two subscales: the Classical Prejudice (7 items, e.g., “Immigrants are generally not very intelligent”; α at T1 = 0.80, α at T2 = 0.81) and the Modern Prejudice (8 items, e.g., “Immigrants are getting too demanding in the push for equal rights”; α at T1 = 0.79, α at T2 = 0.73). The items were rated on a 5-point Likert-type scale ranging from “strongly disagree” (1) to “strongly agree” (5). 

The Social Dominance Orientation Scale [37,55] includes 7 items (e.g., “Inferior groups should stay in their place”; α at T1 = 0.83, α at T2 = 0.84). The items were rated on a 5-point point Likert-type scale ranging from “strongly disagree” (0) to “strongly agree” (4).

The Sociopolitical Control Scale [24,56] includes 8 items (α at T1 = 0.74, α at T2 = 0.77) measuring Leadership Competence (e.g., “I am often a leader in groups”) and 9 items (α at T1 = 0.62, α at T2 = 0.65) measuring Policy Control (e.g., “People like me have the ability to participate effectively in political activities and decision making”). The items were rated on a 4-point point Likert-type scale ranging from “totally false” (1) to “totally true” (4).

Social Trust was assessed using two items [57]: (a) “In general, most people can be trusted”, and (b) “Most people are fair and don’t take advantage of you” (r at T1 = 0.45, *p* < 0.001; r at T2 = 0.57, *p* < 0.001). Response options ranged from “strongly disagree” (1) to “strongly agree” (5).

The Trust in Institutions Scale [52]. Respondents indicated their level of confidence in several institutions, such as the government, political parties, the police, the European Union, and the United Nations Organizations (α at T1 = 0.81, α at T2 = 0.78). Responses were provided on 4-point scales ranging from “none at all” (1) to “great deal of confidence” (4).

The Civic Engagement Scale [58] was used to assess respondents’ participation. This instrument contains 10 items (α at T1 = 0.86, α at T2 = 0.85) and measures the frequency with which participants are currently engaging in a series of civic actions (e.g., attending a public meeting or demonstration dealing with political or social issues; working with others to solve community problems) on a 5-point scale ranging from “never” (1) to 5 “very often” (5). 

The Intention of Civic Engagement Scale [58,59] includes 10 items (α at T1 = 0.82, α at T2 = 0.81) to assess respondents’ intention to participate in the future. Participants were asked to rate the likelihood of being engaged in a series of civic actions (e.g., attending a public meeting or demonstration dealing with political or social issues; working with others to solve community problems) in the next 12 months. The items were rated on a 5-point point Likert-type scale ranging from “very unlikely” (1) to 5 “very likely” (5). 

Finally, a brief list of socio-demographic items was included. 

## 3. Results

### 3.1. Descriptive Statistics at T1

As shown in Table 1, sexism expressed by participants at the beginning of the educational project was quite low, although Benevolent Sexism was slightly higher than Hostile Sexism. Decidedly low was Prejudice against LGBT people, while both Classical and Modern Ethnic Prejudice were slightly higher. Not high, but close to the central point of the response scale, was Social Dominance Orientation. The two dimensions of Sociopolitical Control (Leadership Competence and Policy Control) were quite pronounced among participants, as Social Trust was quite high and Trust in Institutions was somewhat lower. Both the effectiveness and Intention of Civic Engagement were quite pronounced at T1. 

### 3.2. T-Tests and Correlations

Paired sample t-tests were performed to assess whether participants’ scores on the investigated variables changed between T1 and T2. As can be seen in Table 1, the results showed that both Hostile and Benevolent Sexism decreased, as did both Classical and Modern Ethnic Prejudice and Social Dominance Orientation. In contrast, prejudice against LGBT people did not change substantially. Among the variables related to engagement, only Trust in Institutions showed a significant increase between T1 and T2. 

We then conducted bivariate linear correlations to test whether Civic Engagement at T1 and at T2 were actually associated with variables related to participation at Time 1 (i.e., Leadership Competence, Policy Control, Social Trust, Trust in Institutions, Intention of Civic Engagement). As shown in Table 2, Civic Engagement at T1 was positively related to Leadership Competence, Policy Control, Trust in Institutions, Intention of Civic Engagement Civic and actual Civic Engagement at T2. The latter was also significantly associated with Policy Control and Intention of Engagement measured at Time 1. In addition, Intention of Civic Engagement correlated positively with Leadership Competence, Policy Control, Social Trust, and Trust in Institutions. 

Finally, we performed bivariate linear correlations to test whether engagement at T1 and at T2 were negatively associated with prejudices and SDO at T2. As can be seen in Table 3, there was no significant relationship between civic engagement at T1 and prejudices and SDO at T2. Each prejudice was positively correlated with the other prejudices, and all prejudices were significantly associated with SDO. 

### 3.3. Path Analysis to Test the Hypothesised Model

Based on the above-reported literature and on the results of correlations, we tested a model hypothesizing that Leadership Competence, Policy Control, Social Trust, Trust in Institutions, and Civic Engagement at T1 would be positively associated with Intention of Civic Engagement, which in turn would positively influence Civic Engagement at T2. After a power analysis using Gpower 3 software with linear multiple regression, a fixed model, an R2 deviation from zero, and an alpha of 0.05 that confirmed the adequacy of our sample size for the analysis, the hypothesized relationships were tested using AMOS 27. 

The model tested did not show good fit indexes, as two relations (i.e., between Leadership Competence and Intention of Civic Engagement, and between Trust in Institutions and Intention of Civic Engagement) were not-significant. We modified the model, removing the not significant paths. The second model was satisfactory, and all the parameters were statistically significant: *χ*^2^(14) = 49.79, *p* < 0.01; *χ*^2^/gdl = 3.55; CFI = 0.94; TLI = 0.93; RMSEA = 0.07. Figure 1 shows the model in graphic form. It explained 20% of the variance in Intention of Civic Engagement and 33% of the variance in Civic Engagement at T2. We found that Policy Control, Social Trust, and Civic Engagement at T1 increased Intention of Civic Engagement (β = 0.37, *p* < 0.001, β = 0.09, *p* < 0.01, β = 0.51, *p* < 0.001, respectively). Intention of Civic Engagement at T1 was positively associated with actual Civic Engagement at T2 (β = 0.56, *p* < 0.001).

## 4. Discussion 

The purpose of the current longitudinal study was to examine the effects of an educational project aimed at reducing prejudice and developing civic engagement among youth. To this end, project participants completed a questionnaire before beginning and after completing the educational project. We started with three hypotheses.

Regarding hypothesis H1, that participation in the project could reduce prejudice and ideologies related to Social Dominance Orientation, the data showed that participation in the project did indeed reduce both sexist attitudes and ethnic prejudice. The only prejudice that did not change was prejudice against LGBT people. We can suppose that this may be due to the topics explicitly addressed by the training course. Indeed, if some activities were clearly focused on gender and racial issues, no specific attention was paid to sexual orientation. Moreover, prejudice towards LGBT people was not high even at the beginning of the project, consistent with the general trend that it is gradually decreasing among young people [60]. Another result from the comparison between responses at T1 and T2 was that youth exhibited lower levels of social dominance orientation at the end of the project. Thus, with the exception of LGBT prejudice, hypothesis H1 found confirmation.

Hypothesis H2, i.e., that constructs related to participation in previous studies (e.g., [15,16]) should increase after involvement in the project, was partially confirmed. Only trust in institutions increased significantly, while leadership competence, policy control, and social trust did not.

These findings are consistent with scholars’ reasoning that educational interventions aimed at identifying social inequalities and developing a critical analysis of relationships between advantaged and disadvantaged groups primarily promote the reduction of prejudice [30,31]. These issues were addressed during the peer education training. Furthermore, the real or imagined contact [28] that participants experienced during the travels may have contributed to the reduction of prejudice. Other dimensions, such as leadership competence, policy control, and social trust, may have been addressed less directly during the project and did not change significantly. The attention paid to the law and the role of European institutions during the classroom activities may have increased participants’ trust. 

Hypothesis H3, which refers to the constructs that favor the development of commitment, was partially confirmed. Indeed, the most important constructs for sustaining the intention to participate, which actually leads to participation, are policy control, social trust, and current civic engagement, while leadership competence and trust in the institution have no influence.

These findings are consistent with previous studies with adolescents, similar to our work, which emphasized that trust in the social context can foster engagement in the local community and the intention to contribute to social well-being [2,16]. Furthermore, as noted, participation is also based on individual beliefs regarding the ability to act effectively in social and political contexts, i.e., policy control. These skills could be developed and maintained through educational interventions that specifically focus on the ability to effectively influence social and political contexts. Another dimension that affects intention to participate is social trust, which cannot be “taught” but can be fostered by offering positive role models and creating a non-judgmental learning situation. This kind of educational experience can allow young people to experience a sense of safety that is at the basis of social trust [16]. Finally, current civic engagement is an important predictor of the intention to continue participating. Indeed, actual participation behaviors have already overcome barriers to involvement [61,62]. Furthermore, previous participation occasions may have produced some positive experiences that feed the desire to repeat them [63,64].

## 5. Limitations and Conclusions

The present research work has some limitations. First, participants were not numerous at Time 2, as a high attrition rate was registered between T1 and T2. In addition, the participants were only from North Italian regions, limiting the generalizability of the results.

Another weakness of the study is that a follow-up questionnaire a few months later was not possible; this could have allowed to assess the maintenance of changes or to investigate other long-term effects.

Despite these limitations, the longitudinal design of the study provided an opportunity to assess factors affecting youth civic engagement and provided some clues for future research and application. In particular, it would be interesting to explore in more depth the relationships between intention to engage in civic participation and prejudice and social dominance orientation; in our results, they showed no correlation, in contrast to Banks et al. [65] and Ang et al. [66], respectively.

Moreover, the results of the study provide some suggestions for the development of future educational interventions, both in terms of content and process. With the goal of promoting youth civic participation, a particular focus could be on improving participants’ perceptions of their ability to effectively interact with and influence social and institutional contexts, or leadership competence. Furthermore, fostering social trust can be important; this can be facilitated and developed by creating a non-judgmental and friendly educational environment. In this direction, a non-formal approach such as peer education, based on partnership and the active involvement of young people, could be useful. The role of adult educators is also crucial: they should offer good and “solid” models and solicit critical thinking without judging. The development of leadership competence, policy control, and social trust can also be supported by educational methods that combine learning by doing and direct experience, similar to the method proposed in this project. However, these dimensions are complex, and perhaps more time is needed to develop them than the duration of the project.

Another issue is the context in which the educational proposal is provided. The school can play a key role because it can (or should) reach all young people, as we said before [13,47]. Moreover, it is one of the first institutions that young people come into contact with; if it provides reliable proposals, it can contribute to the development of trust in institutions. However, non-formal educational institutions, such as the third-sector associations that carried out the project of this study, are no less important. Indeed, they are often in a position to provide innovative and experiential stimuli and to create the conditions for bottom-up youth initiatives, which are important for activating and sustaining youth civic participation [67].

## Figures and Tables

**Figure 1 behavsci-13-00650-f001:**
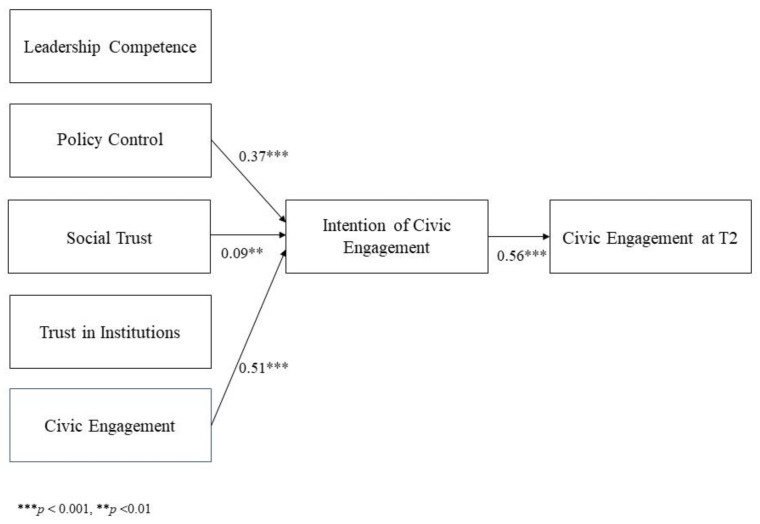
Standardized path coefficients of the final model. Only significant paths are reported.

**Table 1 behavsci-13-00650-t001:** Mean scores (M) and standard deviations (SD) at Time 1 and Time 2 and Paired sample t-tests (t values) to test differences between Time 1 and Time 2.

		M (SD)		
	Min–Max	Time 1	Time 2	t
Hostile Sexism	0–5	1.44 (1.08)	1.18 (1.08)	4.21 ***
Benevolent Sexism	0–5	2.09 (1.18)	1.88 (1.17)	3.43 ***
LGBT Prejudice	1–5 ^1^	4.70 (0.58)	4.73 (0.56)	−1.03
Classical Ethnic Prejudice	1–5	2.01 (0.59)	1.91 (0.59)	2.47 *
Modern Ethnic Prejudice	1–5	2.15 (0.63)	1.99 (0.60)	3.80 ***
Social Dominance Orientation	0–4	1.73 (0.67)	1.60 (0.61)	3.01 **
Leadership Competence	1–4	2.80 (0.55)	2.83 (0.54)	−0.92
Policy Control	1–4	2.66 (0.46)	2.63 (0.48)	0.94
Social Trust	1–5	2.99 (0.81)	3.03 (0.90)	−0.52
Trust in Institutions	1–5	2.29 (0.43)	2.41 (0.42)	−3.48 ***
Civic Engagement	1–5	2.79 (0.72)	2.81 (0.68)	−0.31
Intention of Civic Engagement	1–5	2.88 (0.64)	2.93 (0.66)	−0.90

*** *p* < 0.001; ** *p* < 0.01; * *p* < 0.05. ^1^ While for Hostile and Benevolent Sexism and for Classical and Modern Ethnic Prejudice the response options range from “strongly disagree” to “strongly agree”, for LGBT Prejudice they range from “strongly agree” to “strongly disagree”.

**Table 2 behavsci-13-00650-t002:** Zero-order correlations between Civic Engagement at Time 1 and 2, Sociopolitical Control and Trust at Time 1.

	1	2	3	4	5	6
1. Civic Engagement T1						
2. Civic Engagement T2	0.52 **					
3. Leadership Competence T1	0.15 **	0.05				
4. Policy Control T1	0.36 **	0.23 **	0.28 **			
5. Social Trust T1	0.06	0.04	0.00	0.08		
6. Trust in Institutions T1	0.10 *	0.04	0.12 *	0.21 **	0.24 **	
7. Intention of Civic Engagement T1	0.67 **	0.54 **	0.20 **	0.47 **	0.18 **	0.19 **

** *p* < 0.001, * *p* < 0.05.

**Table 3 behavsci-13-00650-t003:** Zero-order correlations between Civic Engagement at Time 1 and 2, prejudices and Social Dominance Orientation at Time 2.

	1	2	3	4	5	6	7
1. Civic Engagement T1							
2. Civic Engagement T2	0.52 **						
3. Hostile Sexism T2	−0.05	0.04					
4. Benevolent Sexism T2	−0.07	0.11	0.50 **				
5. LGBT Prejudice T2	−0.05	−0.10	0.37 **	0.34 **			
6. Classical Ethnic Prejudice T2	−0.17	−0.11	0.47 **	0.32 **	0.46 **		
7. Modern Ethnic Prejudice T2	−0.11	−0.18	0.49 **	0.32 **	0.50 **	0.57 **	
8. Social Dominance Orient. T2	−0.15	−0.15	0.52 **	0.32 **	0.50 **	0.54 **	0.70 **

** *p* < 0.001.

## Data Availability

The data that support the findings of this study are available from the corresponding author, upon reasonable request.

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
