# Peer review of "Educating Youth to Civic Engagement for Social Justice: Evaluation of a Secondary School Project"

_behavsci, 2023, doi:10.3390/bs13080650_

Round 1

Reviewer 1 Report

The reviewed manuscript topic, educating youth engagement in civic actions for social justice, represents a relevant issue nowadays. The study presents a way to develop a set of attitudes and skills crucial in any society or culture that aims at social balance, social cohesion, a strong sense of community, and peace. One of its strengths is the longitudinal design used to evaluate the effectiveness of an educational intervention to increase high-school students' civic engagement for social justice. 

I hope my comments can assist in the further development of this manuscript.

1.    The Introduction section is well written but the project section may benefit from a more detailed description such as the number of training hours per week, how many themes or topics were approached, or any other detail related to the internal structure of the intervention

2.    Method: Please provide additional information about the participant recruitment method and the demographic details of your participants to help assess the generalizability of your findings. Please consider adding such information as a) a description of any inclusion/exclusion criteria that were applied to participant recruitment, b) more demographic details, c) a statement as to whether your sample can be considered representative of a larger population, d) a description of how participants were recruited, e) descriptions of why authors use a convenience sampling method in this study, and f) explain how this sample size is sufficient and reliable for analysis.

3.    Regarding the convergent validity, it would be preferred to report the composite reliability (CR) and the average variance extracted (AVE) of the measurement model.

4.    Figure 1: This figure is commonly known as a path analysis model. If it is a structural equation model, it must have a measurement model and latent variables. Is it possible for the authors to show a full figure of the SEM of this research that consists of the structural model and measurement model?

5.    On line 389, please check if the term attrition is more accurate instead of high mortality.

Author Response

Thank you for your useful suggestions. Please see the attached file for the point by point answers.

the authors

Reviewer 2 Report

Recommendations to author(s)

The aim of the article complies with the scope of the journal and gives valuable new knowledge for understanding the effects of youth civic engagement and for designing and implementing civic education interventions for young people, especially in school settings.

The following changes or additions might help to slightly improve the article:

1.       Section 1.3. The project - Consider presenting more information about the educational project/intervention, especially in relation to the topics that were addressed. It would help to understand the issue of the topic that the author(s) addressed in the discussion (Section 4 Discussion, page 10, lines 345-351)

2.       In Section 2.2. all the relevant measures were addressed. However, in order to follow the result and discussion more easily there is some missing information in relation to the meaning/interpretation of the total results. Therefore, consider presenting some information about the total result meaning/interpretation, for example, does the result on The LGBT Prejudice Scale means having more or less prejudice? I assume that the items were rated also on the Likert scale from strongly disagree (0) to strongly agrees (5), or not?

3.       In line with the previous comment consider giving some short descriptive information about the T1 sample in relation to the used/addressed measures. It seems the sample from the beginning was “low” in all measures, which was discussed only considering LGBT prejudice

4.       Section 4. Discussion, page 11, lines 372-373: The discussion would benefit from more explanation if the previous studies explored the trust among youth or the general population, and if there are some differences considering different age groups/developmental phases since this survey is focused on youth.

5.       Section 5 Limitations and conclusion, line 389: What does it mean by “high mortality”, please explain or refrase.

Author Response

(The authors gave the same response as above.)
